# JAEGER: Joint 3D Audio-Visual Grounding and Reasoning in Simulated Physical Environments

**Zhan Liu** [1 2]  **Changli Tang** [1]  **Yuxin Wang** [3 2]  **Zhiyuan Zhu** [4 2]  **Youjun Chen** [5 2]  **Yiwen Shao** [2]  **Tianzi Wang** [2]
**Lei Ke** [2]  **Zengrui Jin** [1]  **Chao Zhang** [1]

## Abstract

Current audio-visual large language models (AV-LLMs) are predominantly restricted to 2D perception, relying on RGB video and monaural audio. This design choice introduces a fundamental dimensionality mismatch that precludes reliable source localization and spatial reasoning in complex 3D environments. We address this limitation by presenting JAEGER, a framework that extends AV-LLMs to 3D space, to enable joint spatial grounding and reasoning through the integration of RGB-D observations and multi-channel first-order ambisonics. A core contribution of our work is the neural intensity vector (Neural IV), a learned spatial audio representation that encodes robust directional cues to enhance direction-of-arrival estimation, even in adverse acoustic scenarios with overlapping sources. To facilitate large-scale training and systematic evaluation, we propose SpatialSceneQA, a benchmark of 61k instruction-tuning samples curated from simulated physical environments. Extensive experiments demonstrate that our approach consistently surpasses 2D-centric baselines across diverse spatial perception and reasoning tasks, underscoring the necessity of explicit 3D modelling for advancing AI in physical environments. Our source code, pre-trained model checkpoints, and datasets are available at https://github.com/liuzhan22/JAEGER.

## 1. Introduction

Despite the rapid evolution of audio-visual large language models (AV-LLMs), most existing systems continue to rely

[1]Tsinghua University [2]Tencent AI Lab [3]HKUST [4]Zhejiang University [5]The Chinese University of Hong Kong. Correspondence to: Chao Zhang <cz277@tsinghua.edu.cn>.

*Proceedings of the 43rd International Conference on Machine Learning*, Seoul, South Korea. PMLR 306, 2026. Copyright 2026 by the author(s).

on RGB video and monaural audio (Xu et al., 2025a;b; Tang et al., 2025; Cheng et al., 2024), a design choice that fundamentally constrains their ability to perceive and reason about the three-dimensional (3D) physical world. Although 3D grounding has recently attracted significant attention, current research remains fragmented, with most approaches addressing spatial cues from vision or audio in isolation. In the visual domain, recent advances extend a 2D visual LLM by incorporating RGB-D inputs and chain-of-thought reasoning, leading to improved 3D grounding and spatial relationship understanding (Wang et al., 2025). In parallel, auditory approaches leverage binaural spatial encoders (Zheng et al., 2024) or intensity-vector representations (Tang et al., 2024) to enable relative sound source localization. However, a unified paradigm for comprehensive 3D audio-visual scene understanding remains largely unexplored. Early multimodal efforts, such as Hear You Are (Ryu et al., 2026), pair spatial audio with panoramic RGB image and assume a single active source per scene, which precludes evaluating overlap-robust localization and depth-aware 3D grounding. Similarly, while SAVVY (Chen et al., 2025) integrates RGB-D perception with multi-channel audio, it relies on a cascaded pipeline that depends on traditional signal processing for sound source localization. This modular design hinders end-to-end learning and prevents the AV-LLM from performing fully integrated spatial reasoning, underscoring the need for a more unified, learned approach.

In this paper, we introduce JAEGER, a **j**oint 3D **a**udio-visual r**e**asoning and **g**rounding method in simulated physical **envi**ronments, an end-to-end framework that extends 2D audio-visual large language models (AV-LLMs) to 3D settings by explicitly modeling visual depth and multi-channel spatial audio. Initialized from Qwen2.5-Omni (Xu et al., 2025a) and efficiently adapted via low-rank adaptation (LoRA) (Hu et al., 2022), JAEGER integrates an RGB-D visual stream augmented with depth-projected 3D positional encodings together with a dual-path audio stream, in which semantic content extracted from the omnidirectional first-order ambisonics (FOA) channel is disentangled from spatial directional cues. To further enhance azimuth perception in reverberant environments and under overlapping sound sources, we propose a novel neural intensity vec-

tor (Neural IV) approach, which is a learnable FOA-based representation that encodes robust directional information for improved sound localization and cross-modal alignment.

To support instruction tuning and systematic evaluation, we construct SpatialSceneQA, a benchmark comprising 61k high-fidelity RGB-D scenes paired with spatial audio and fine-grained 3D annotations. The dataset covers a diverse set of tasks, including single- and overlapping-source direction-of-arrival (DoA) estimation, 3D visual grounding of sound-emitting objects, and multi-speaker audio-visual matching. Experimental results show that JAEGER achieves a median angular error (MAE) of 2.21° for single sources and 4.11° under overlapping conditions, while showing strong generalization across varied source configurations. Leveraging explicit depth cues, the model attains a 3D intersection over union (IoU) of 0.32 with a median localization error of 0.16 meter (m). When FOA-based spatial cues are jointly modeled with RGB-D perception, JAEGER further achieves 99.2% accuracy on joint audio-visual reasoning in simulated multi-speaker physical environments.

In summary, our main contributions are as follows:

- We introduce joint grounding and reasoning with spatial audio and RGB-D geometry, which exposes a fundamental modality gap for current AV-LLMs even after fine-tuning. We adapt an AV-LLM with depth-aware visual encoding and FOA spatial cues for end-to-end DoA estimation, 3D box grounding, and multi-speaker matching.
- We present SPATIALSCENEQA, a 61k-sample dataset that pairs 3D RGB-D images with 3D 4-channel FOA audio and dense object-level spatial annotations. To the best of our knowledge, it is the first spatial audio-visual benchmark with degree-level azimuth and elevation supervision, enabling precise localization, 3D grounding, and multi-speaker matching under source overlap.
- We propose the *Neural Intensity Vector*, a learnable FOA spatial representation that replaces STFT-based intensity features with a neural encoder, yielding more stable azimuth cues under reverberation and overlapping sources and improving cross-scenario generalization.

## 2. Related Work

### 2.1. Spatial Audio Understanding

Spatial audio understanding seeks to infer the spatial configuration of acoustic events from multi-channel recordings, including binaural signals, FOA, and microphone arrays. Early work has primarily focused on sound event localization and detection (Adavanne et al., 2018; Cao et al., 2021; Shimada et al., 2021; 2022), where models jointly estimate event activity and coarse spatial attributes. Recent studies have begun to integrate spatial audio perception with large

language models (LLMs), enabling instruction following and higher-level reasoning grounded in spatial acoustic cues.

A major design choice is the spatial representation. Binaural modeling leverages interaural time and level differences and head-related transfer functions (HRTFs) induced cues, and recent geometry-aware encoders improve the use of such signals (Zheng et al., 2024; Biswas et al., 2026). However, binaural cues are tightly coupled to recording geometry and the underlying HRTFs, which hinders generalization across devices. FOA has a hardware-agnostic alternative by encoding spatial information in the B-format. (Devnani et al., 2024) aligns FOA-based spatial embeddings with semantics via contrastive learning, while (Tang et al., 2024) injects intensity vector cues into an auditory LLM to support localization and localization-conditioned speech processing.

Progress in this area is further constrained by data availability. Real-world multi-channel datasets with dense spatial annotations remain scarce, with STARSS23 (Shimada et al., 2023) serving as a representative benchmark. However, STARSS23 pairs multi-channel audio with RGB-only paranomic video, and does not provide depth aligned to the visual stream. This limitation has motivated simulation-driven data generation, where configurable simulators such as SoundSpaces 2.0 (Chen et al., 2022) enable scalable synthesis of reverberant or anechoic audio-visual data with controllable scene geometry and precise ground-truth spatial annotations.

### 2.2. Visual 3D Object Grounding

Three-dimensional visual grounding and spatial reasoning have become increasingly important for modern vision–language models (Huang et al., 2024; Chen et al., 2024b; Bai et al., 2025; Guo et al., 2025; Zhu et al., 2025b) and AV-LLMs (Senocak et al., 2023; Chowdhury et al., 2024; Li et al., 2024), as many downstream tasks demand object-level localization and relational reasoning beyond two-dimensional captioning. Existing approaches primarily differ along two dimensions: (i) how 3D geometric information is incorporated and (ii) whether the model natively predicts 3D outputs.

A first line of work conditions models on explicit 3D structure, such as point clouds or object-centric 3D tokens, to impose direct geometric constraints (Huang et al., 2024; Mao et al., 2025). An alternative relies on multi-view video or RGB-D streams as implicit 3D supervision (Qi et al., 2026; Zheng et al., 2025; Chen et al., 2024a; Zhu et al., 2025a). (Zheng et al., 2025) projects depth into global coordinates and injects 3D positional encodings into video representations, improving spatial awareness through geometry-aligned tokens. Similarly, RGB-D-based approaches incorporate depth-derived cues into visual tokens to strengthen geometric grounding (Chen et al., 2024a; Zhu et al., 2025a).

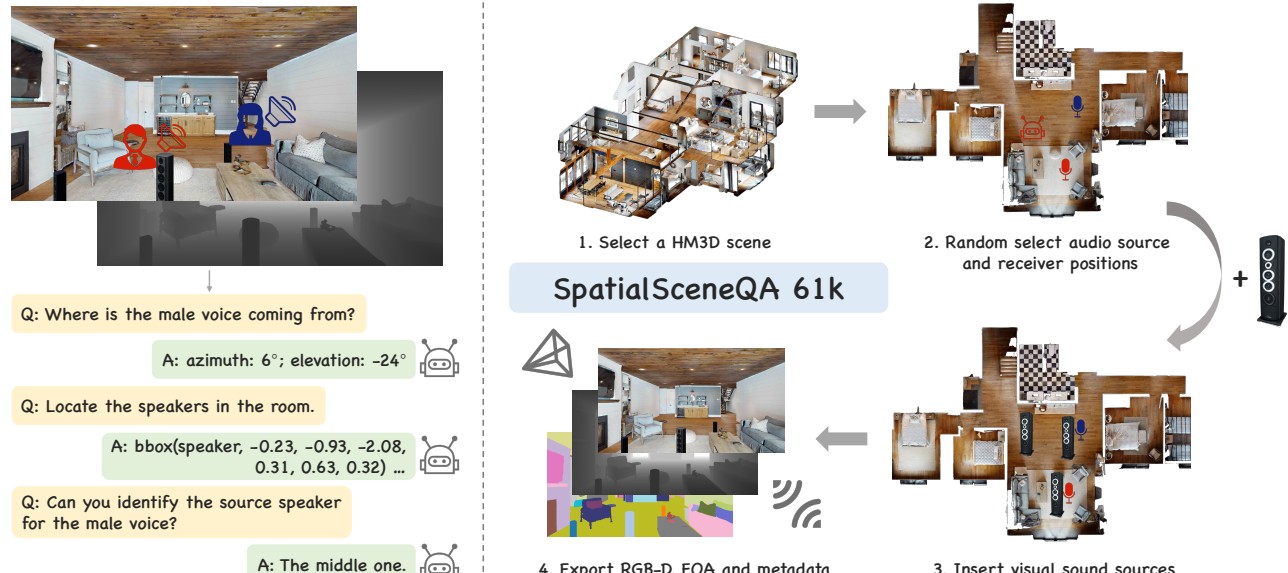

*Figure 1.* **Overview of the SPATIALSCENEQA 61k dataset.. Left:** Example question-answer pairs demonstrating diverse spatial tasks, including sound source localization (azimuth/elevation), visual grounding (bounding boxes), and overlapping sound source identification. **Right:** The data synthesis pipeline leveraging *Habitat-Sim* and *SoundSpaces 2.0*. The process consists of four stages: (1) selecting an HM3D scene, (2) sampling random source and receiver poses, (3) inserting 3D visual sound sources (e.g., speakers generated by Hunyuan3D-1.0), and (4) exporting synchronized RGB-D frames, FOA audio, and semantic and camera intrinsic/extrinsic metadata.

Several approaches rely on auxiliary components to obtain reliable 3D signals, such as external 3D segmenters for candidate proposal generation or specialized decoders for 3D grounding (Zheng et al., 2025; Zhu et al., 2025a). More recently, N3D-VLM (Wang et al., 2025) advances a more native paradigm by explicitly reasoning about and directly predicting 3D bounding boxes within the model.

### 2.3. AV-LLMs for 3D Understanding

Recent AV-LLMs show strong multimodal understanding (Cheng et al., 2024; Xu et al., 2025a;b; Tang et al., 2025), but are largely trained on RGB video and monaural audio, leaving spatial structure and directional acoustics implicit. This limitation motivates a line of work that augments AV-LLMs with explicit spatial cues for precise localization and geometric reasoning. One representative approach builds modular pipelines that estimate geometry and spatial acoustics using specialized components, while delegating higher-level reasoning to an LLM. (Chen et al., 2025) introduces a benchmark for dynamic 3D audio-visual spatial reasoning with synchronized spatial audio and proposes a training-free pipeline that estimates object trajectories and constructs a global spatial map to answer queries. Complementary simulation-based efforts, such as Hear You Are (Ryu et al., 2026), exploit panoramic imagery and binaural audio cues for audio-visual spatial reasoning, but continue to rely on RGB-only visual inputs, leaving explicit depth information unmodeled.

In contrast, our work adapts an existing AV-LLM to directly perceive and reason about spatial cues by incorporating RGB-D observations and FOA-based multi-channel audio in an end-to-end manner. We introduce a learned spatial audio representation, the Neural IV, and construct a large-scale instruction-tuning dataset, SpatialSceneQA, to jointly support spatial perception, grounding, and language-based reasoning within a unified framework.

## 3. SpatialSceneQA

Training a 3D AV-LLM demands large-scale supervision with tightly synchronized (i) metric geometry (RGB-D with camera calibration), (ii) multi-channel spatial audio, and (iii) 3D object-level annotations in a consistent reference frame. To this end, we introduce SPATIALSCENEQA 61K, a synthetically generated dataset for high-fidelity 3D audio-visual instruction tuning. Each sample contains synchronized RGB-D renderings, 4-channel FOA, and precise 3D annotations, enabling supervision for both spatial perception and audio-visual spatial reasoning.

### 3.1. Data Simulation Pipeline

We synthesize SPATIALSCENEQA using *SoundSpaces 2.0* (Chen et al., 2022), which supports continuous, geometry-aware acoustic rendering in scanned 3D environments. The pipeline generates synchronized FOA audio and RGB-D observations, together with calibrated camera metadata and

*Table 1.* Statistics of the SPATIALSCENEQA 61K dataset. Tasks are grouped into "Perception" and "Reasoning categories. **#Sources** denotes active audio sources for A–B and inserted loudspeaker assets for C–E. Regarding definitions of different types of tasks, **A–B** estimate source azimuth and elevation; **C** predicts the precise Bounding Box of the inserted speaker; **D–E** identify the correspondence between a target audio speaker and a visual loudspeaker based on spatial audio-visual cues.

| Task Type | #Sources | #Samples | Question & Answer Example |
|---|---|---|---|
| *Perception* | | | |
| **A: Single-Source Audio DoA** | 1 | 32K | **Q:** Based on the audio cues, please output the precise azimuth and elevation. **A:** azimuth: -7; elevation: -22 |
| **B: Overlap-Source Audio DoA** | 2 | 30K | **Q:** Where is the female voice coming from? Output azimuth and elevation. **A:** azimuth: 28; elevation: -15 |
| **C: Visual Grounding** | 1 2 3 | 17K 34K 9K | **Q:** Identify the 3D box for the speaker in this scene. **A:** `bbox_0 = Bbox`$(\text{speaker}, 0.14, -0.48, -1.15, 0.33, 0.88, 0.32)$ |
| *Reasoning* | | | |
| **D: Single-Source Multi-speakers Matching** | 2 3 | 10K 4K | **Q:** Determine which speaker corresponds to the audio source. **A:** Left |
| **E: Overlap-Source Multi-speakers Matching** | 2 3 | 24K 5K | **Q:** Can you tell which speaker the male voice originates from? **A:** Center |

3D ground truth, as presented in Fig. 1.

**Audio Source Selection.** We use dry monaural speech from *LibriSpeech* (Panayotov et al., 2015), which contains read speech recordings with aligned transcripts and speaker metadata. To prevent data leakage, we construct the training set from train-clean-100, use dev-clean for validation, and reserve test-clean for evaluation. For tasks involving speaker identity attributes, the provided gender labels are used to specify the target speaker in the instruction.

**Acoustic Simulation.** *SoundSpaces 2.0* renders multi-channel spatial audio by simulating room impulse responses (RIRs) over realistic 3D meshes with material-dependent acoustics. It employs bidirectional path tracing (Cao et al., 2016) to model direct sound and higher-order effects, including reflections, transmission, diffraction, and air absorption. Given a monaural source signal $A^{(s)}(t)$ emitted from position $\mathbf{s} \in \mathbb{R}^3$ and a receiver at position $\mathbf{r} \in \mathbb{R}^3$ with heading $\theta \in \mathbb{R}$, the received FOA signal on channel $c$ is

$$A_c^{(r)}(t) = \big(R_c(\cdot; \mathbf{s}, \mathbf{r}, \theta) * A^{(s)}\big)(t), \tag{1}$$

where $*$ denotes convolution over time, $t \in [1, T]$ and $c \in \{0, 1, 2, 3\}$ indexes time steps and FOA channels respectively, $R_c(t, \mathbf{s}, \mathbf{r}, \theta)$ stands for the channel-specific RIR. To reduce degenerate cases caused by fully occluded direct paths, we sample source–receiver pairs within the same room. Consequently, we sample a navigable receiver pose, then sample a source position within 1–4 m, enforcing a 0.5 m clearance from obstacles. Pairs were retained only when the geodesic distance was less than twice the Euclidean distance, ensuring navigability and connectivity.

**Visual Scene Construction.** We use *Habitat-Sim* (Szot et al., 2021) for rendering RGB-D observations aligned to the audio. Scenes are drawn from the semantically anno-tated subset of the Habitat-Matterport 3D Dataset (HM3D) (Ramakrishnan et al., 2021), enabling precise instance-level ground truth. Scene-based data partitioning was employed, with 130/15/36 scenes for train/val/test, respectively. The synchronized RGB, depth, semantic instance masks, and calibrated camera metadata, along with the receiver pose and the ground-truth source poses, are exported.

**Synthetic 3D Object Generation.** Static HM3D scans contain limited diversity of visually salient sound-emitting objects. To increase appearance and geometry diversity while keeping metric annotations exact, synthesized loud-speaker meshes were generated by *Hunyuan3D-1.0* (Yang et al., 2024) and further inserted. We generate 120 distinct floor-standing loudspeaker models and split them into 96/12/12 for train/validation/test to evaluate generalization to unseen geometries. A visibility constraint is enforced by requiring each target loudspeaker to occupy at least 500 pixels in the semantic frame (rendered at $1920 \times 1080$), filtering out heavily occluded or distant instances.

### 3.2. Creation of QA Dataset

We convert each simulated scene into instruction-answer pairs as shown in Table 1, spanning two modules: *Perception* and *Reasoning*.

**Perception Tasks (Task A–C).** Tasks A–B supervise acoustic localization by regressing DoA in spherical coordinates (azimuth, elevation) under both single-source task (A) and overlapping-source task (B) settings. We express angles in degrees under a right-handed camera coordinate system with $x$ pointing right, $y$ up, and $z$ pointing backwards; azimuth is measured on the horizontal plane with $0°$ facing forward, and elevation is measured from the horizontal plane.

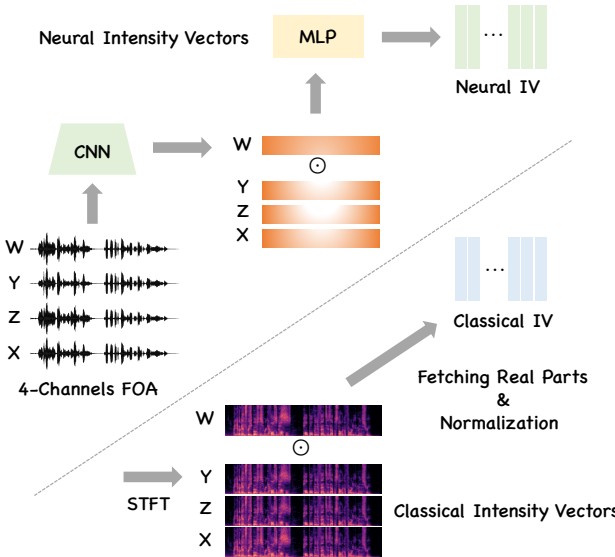

*Figure 2.* Comparisons between Classical IV and Neural IV.

Task C targets metric 3D visual grounding of sound-emitting objects. Following N3D-VLM (Wang et al., 2025), we represent each 3D bounding box as a structured sequence bbox$(c, x, y, z, s_x, s_y, s_z)$, where $c$ is the category label, $(x, y, z)$ is the box center in the camera coordinate system, and $(s_x, s_y, s_z)$ are axis-aligned box dimensions. To avoid learning shortcuts from domain gaps, $\{1, 2, 3\}$ loudspeaker instances were randomly inserted per scene and require the model to output 3D boxes for all visible loudspeakers, enforcing explicit geometric grounding.

**Reasoning Tasks (Task D–E).** Tasks D–E evaluate cross-modal spatial correspondence: given spatial audio and an RGB-D observation containing multiple candidate loud-speakers, the model must identify which visual loudspeaker matches a target speech source. Task D uses a single active source, while Task E introduces overlapping sources and specifies the target by gender. To avoid the appearance of trivial solutions, at least two visible candidate loudspeakers in every reasoning sample are guaranteed. Solving these tasks requires integrating (i) directional cues from FOA, (ii) 3D localization of candidate loudspeakers, and (iii) metric alignment between acoustic DoA and visual geometry, thereby directly probing audio-visual embedding alignment and multi-step spatial reasoning.

## 4. Method

### 4.1. Intensity Vector for Audio Localization

We extract spatial cues from FOA using the intensity vector (Classical IV) (Yasuda et al., 2020). Following (Tang et al., 2024), we compute Classical IV features and use them to augment semantic audio representations with directional

information. Let $F_W$ denote the complex spectrogram of the omnidirectional FOA channel $W$, and let $F_C$ denote the complex spectrogram of the directional channels $C \in \{X, Y, Z\}$. The cross-spectrum for each axis is obtained as:

$$I'_C = F_W^* \odot F_C, \qquad (2)$$

where $\odot$ denotes the Hadamard product and $F_W^*$ is the complex conjugate of $F_W$. The active intensity feature is then constructed by concatenating the real parts:

$$\mathbf{I} = \text{Concat}_{C \in \{X, Y, Z\}} \left( \mathcal{R}(I'_C) \right), \qquad (3)$$

Finally, we apply $L_2$-normalized $\mathbf{I}/\|\mathbf{I}\|_2$ to ensure numerical stability during training.

### 4.2. Neural Intensity Vector

Classical IV rely on fixed STFT-based signal processing, which often produces suboptimal representations in highly reverberant or overlapping-source environments. To address this limitation, here we introduce the Neural IV, a data-driven spatial encoder that learns to extract robust geometric cues directly from raw FOA waveforms.

As shown in Fig. 2, Neural IV replaces the STFT transformation with a learnable CNN backbone. We adopt the data2vec (Baevski et al., 2022) frontend to ensure 50Hz frame rate, utilizing a 7-layer 1D-CNN with kernel sizes of $(10, 3, 3, 3, 3, 2, 2)$ and strides of $(5, 2, 2, 2, 2, 2, 2)$. Let $f_W$ and $f_C$ denote the CNN-encoded latent features corresponding to the omnidirectional and directional channels $(C \in \{X, Y, Z\})$, respectively. We generalize the physical principle of acoustic intensity to the latent space by computing the element-wise product of the omnidirectional $(f_W)$ and directional $(f_C)$ features:

$$h_C = f_W \odot f_C, \quad \text{where } C \in \{X, Y, Z\}. \qquad (4)$$

These interaction features are then concatenated to form a unified spatial representation $\mathbf{H}'$:

$$\mathbf{H}' = \text{Concat}(h_X, h_Y, h_Z). \qquad (5)$$

Finally, we use a multilayer perceptron (MLP) to project these features into the final robust spatial embedding $\mathbf{v}_{\text{NIV}}$:

$$\mathbf{v}_{\text{NIV}} = \text{Linear}(\text{ReLU}(\text{Linear}(\mathbf{H}'))). \qquad (6)$$

### 4.3. 3D-aware Visual Encoding

To empower the visual encoder with 3D awareness, we employ a widely used depth embedding strategy (Zheng et al., 2025; Wang et al., 2025). Specifically, we inject sinusoidal encodings of downsampled metric coordinates into the visual embeddings.

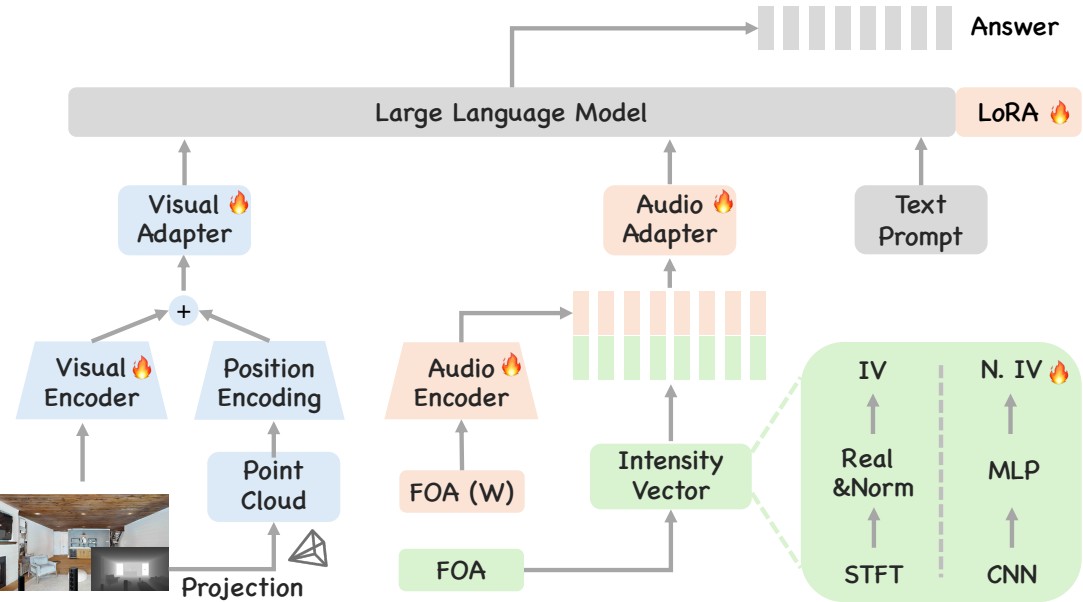

*Figure 3.* **Overview of the JAEGER Architecture.** The framework processes RGB-D and FOA inputs. **(1) Visual Stream:** RGB features are fused with 3D-aware positional encodings derived from depth-projected Point Clouds. **(2) Audio Stream:** Semantic features are extracted from the omnidirectional channel (FOA W). For spatial cues, we compare Classical IV with Neural IV (N. IV). Specifically, IV derives features via STFT followed by fetching real parts and normalization, whereas Neural IV extracts geometric features from raw waveforms using channel-wise 1D-CNNs followed by an MLP.

**Metric Point Cloud Reconstruction.** Given an RGB image $I \in \mathbb{R}^{H \times W \times 3}$ and its aligned depth map $D \in \mathbb{R}^{H \times W}$, we reconstruct the metric point cloud $P \in \mathbb{R}^{H \times W \times 3}$. For each pixel location $(u, v)$ with depth value $D_{uv}$, the corresponding 3D coordinate $P_{uv}$ is obtained via back-projection:

$$P_{uv} = D_{uv} \cdot \begin{bmatrix} f_x & 0 & c_x \\ 0 & f_y & c_y \\ 0 & 0 & 1 \end{bmatrix}^{-1} \begin{bmatrix} u \\ v \\ 1 \end{bmatrix}, \qquad (7)$$

where $(u, v)$ denotes the 2D pixel coordinates, and $f_x, f_y$ and $(c_x, c_y)$ represent the camera's focal lengths and principal point, respectively.

**3D-aware Positional Encoding.** We align the dense geometry $P \in \mathbb{R}^{H \times W \times 3}$ with the spatial resolution of visual features $F_{\text{visual}} \in \mathbb{R}^{h \times w \times c}$ via adaptive average pooling. Each coordinate component $\alpha \in \{x, y, z\}$ of the aligned geometry is mapped to $\lfloor \frac{c}{3} \rfloor$ channels by sinusoidal functions:

$$\begin{aligned} \text{PE}(\alpha, 2j) &= \sin\left(\frac{\alpha}{10000^{2j/\lfloor \frac{c}{3} \rfloor}}\right), \\ \text{PE}(\alpha, 2j+1) &= \cos\left(\frac{\alpha}{10000^{2j/\lfloor \frac{c}{3} \rfloor}}\right), \end{aligned} \qquad (8)$$

for $j = 0, 1, \ldots, \lfloor \frac{c}{6} \rfloor - 1$. The embeddings for the three axes are concatenated to form $F_{3D} \in \mathbb{R}^{h \times w \times c}$, which is then fused with the visual features via element-wise addition:

$$\tilde{F}_{\text{visual}} = F_{\text{visual}} + F_{3D}. \qquad (9)$$

### 4.4. Overall Model Architecture

As shown in Figure 3, JAEGER unifies multi-modal perception and reasoning through the following core modules:

**Visual Stream.** We fuse RGB semantic tokens with geometric priors derived from our 3D-aware Positional Encoding to explicitly ground visual representations in metric space.

**Audio Stream.** We adopt a dual-path design, extracting semantic content from the omnidirectional channel while capturing spatial cues via either *Classical IV* or *Neural IV*.

**LLM Backbone.** All multi-modal features are aligned via MLP adapters and fed into the LLM. We employ LoRA (Hu et al., 2022) to efficiently fine-tune the model for joint 3D grounding and reasoning.

## 5. Experiments

### 5.1. Implementation Details

**Model Specifications.** To leverage the understanding capabilities of existing multimodal models, we initialize the core components of JAEGER, including the visual encoder, the monaural audio branch, and the LLM decoder, with pre-trained weights from Qwen2.5-Omni (Xu et al., 2025a). The proposed Neural IV is randomly initialized. Since the audio embedding dimension has been changed, the audio adaptor has also been initialized randomly.

*Table 2.* Main results compared with baselines. **Task Definitions & Metrics:** (1) **Audio DoA**: Detection of sound source arrival direction. Metric: Median Angular Error (MAE) in degrees (°). (2) **Overlap Audio DoA**: MAE of a specific audio source's arrival angle when two audio clips overlap. (3) **3D IoU**: Mean Intersection over Union of the predicted 3D bounding box. (4) **Visual Offset**: Median distance deviation of the predicted object center coordinates from the ground truth. (5) **Reasoning 1-speaker**: Localizing one of multiple visible loudspeakers in the scene based on spatial audio cues from a single speaker. (6) **Reasoning 2-speaker**: Identifying a specific loudspeaker among multiple candidates based on the spatial location cues of a target speech in a two-speaker scenario. Reasoning tasks are evaluated by **Accuracy**. "–" indicates the model is incapable of performing the task.

| Model | Modality | Perception | | | | Reasoning | |
|---|---|---|---|---|---|---|---|
| | | Audio DoA ↓ | Overlap DoA ↓ | 3D IoU ↑ | Visual Offset ↓ | 1-speaker ↑ | 2-speaker ↑ |
| Random | N/A | 90° | 90° | 0.00 | ∞ | 45.6 | 47.4 |
| *Open-source omni model* | | | | | | | |
| Qwen2.5-Omni | RGB+Mono | – | – | 0.00 | 2.40 | 35.8 | 44.0 |
| *Open-source specialized models* | | | | | | | |
| BAT | Binaural | **2.16°** | 19.09° | – | – | – | – |
| Qwen3-VL-8B | RGB | – | – | 0.01 | 1.11 | – | – |
| N3D-VLM | RGB-D | – | – | 0.00 | 2.04 | – | – |
| JAEGER (Classical IV) | RGB-D+FOA | 2.95° | 6.44° | 0.32 | 0.16 | 99.5 | 98.6 |
| JAEGER (Neural IV) | RGB-D+FOA | 2.21° | **4.11°** | **0.32** | **0.16** | **99.5** | **99.2** |

**Data and Training Specifications.** We train JAEGER on SPATIALSCENEQA following the split protocols for *LibriSpeech* (Panayotov et al., 2015) and *HM3D* (Ramakrishnan et al., 2021). Each sample provides comprehensive 3D geometric annotations, including camera intrinsics/extrinsics, instance-level semantic IDs, and ground truth metadata for all sound sources and loudspeaker assets.

**Optimization**. We fine-tune specific components selectively: the audio encoder, Neural IV, and projector for Audio DoA (Tasks A–B); the visual encoder and projector for Visual Grounding (Task C); and all modality encoders and projectors for Joint Reasoning (Tasks D–E). Across all tasks, the LLM is optimized via LoRA with $r = 64$, $\alpha = 128$, and a dropout rate of 0.05. All experiments are conducted on NVIDIA A100 (40GB) GPUs. For audio-only localization, we train on eight GPUs with a batch size of 3 for 6k steps. For visual grounding, we utilize 24 GPUs with a minibatch size of 1 for 3k steps. For joint reasoning tasks, we employ twenty-four GPUs with a minibatch size of 1 for 4k steps. We utilize an optimizer with a cosine learning rate scheduler and a linear warm-up of 2,500 steps. The peak learning rate is set to $1 \times 10^{-5}$, with a weight decay of 0.05.

### 5.2. Main Results

**Evaluation Metrics.** We employ distinct metrics tailored to each task type to ensure a comprehensive evaluation. For Sound Source Localization (Tasks A & B), we report the median **DoA Error**, defined as the angular distance between the predicted azimuth/elevation and the ground truth. For Visual Grounding (Task C), we utilize two standard metrics: **3D IoU**, which measures the volumetric overlap between the predicted and ground truth bounding boxes, and **Visual**

**Offset**, calculated as the Euclidean distance ($L_2$ norm) between the centers of the predicted and ground truth objects in 3D spaces. For Joint Audio-Visual Reasoning (Tasks D & E), which are formulated as multi-choice classification problems (*i.e.*, identifying the correct speaker among candidates like Left/Center/Right), we report the **Accuracy**.

**Baselines.** Since JAEGER is the first framework to integrate RGB-D visual streams with 4-channel FOA for joint grounding and reasoning, no direct baseline exists with an identical modality setup. To ensure a rigorous evaluation, we compare three categories of models: **(1)** Specialized models: We evaluate state-of-the-art models in isolated domains. For audio-only localization, we train BAT (Zheng et al., 2024) for 5 epochs on our Task A and B datasets, matching our train duration. To bridge the modality gap, we convert our 4-channel FOA data to binaural audio using the default SoundSpaces 2.0 HRTF. For 3D visual grounding, we assess N3D-VLM (Wang et al., 2025) and Qwen3-VL-8B-Instruct (Bai et al., 2025) under a zero-shot setting. **(2)** Open-source 2D omni model: To assess the necessity of 3D spatial modalities, we test Qwen2.5-Omni (Xu et al., 2025a) in a zero-shot manner. Notably, due to its monaural-only input, Qwen2.5-Omni is inherently incapable of Audio DoA estimation. **(3)** Random: We provide a theoretical random guess baseline to establish the lower bound of performance.

**Analysis.** Regarding isolated perception tasks, JAEGER demonstrates high competency in both audio and visual domains. In audio localization, while our proposed Neural IV variant (MAE 2.21°) achieves comparable precision to the specialized BAT baseline (2.16°) in single-source scenarios, it significantly outperforms BAT in the more challenging overlapping-source task, reducing the MAE from 19.09°

*Table 3.* Cross-evaluation matrix (Median Angular Error in °). We compare **Classical IV** and **Neural IV** by training and testing on matched or mismatched source scenarios.

| Test Set \ Train Set | Single Src. | Overlap Src. |
|---|---|---|
| **Classical IV** | | |
| Single Source | 2.95° | 18.35° |
| Overlap Source | 19.25° | 6.44° |
| **Neural IV** | | |
| Single Source | **2.21°** | **14.91°** |
| Overlap Source | **14.85°** | **4.11°** |

to 4.11°. In terms of 3D visual grounding, JAEGER-3D achieves a 3D IoU of 0.32 and a visual offset of 0.16m, exhibiting strong competitive performance.

A pronounced performance gap emerges on joint reasoning tasks: mainstream 2D AV-LLMs, constrained by monaural audio and the absence of depth perception, fail even after fine-tuning to reliably associate visual objects with acoustic sources in 3D space. In contrast, JAEGER achieves near-perfect accuracy in both single and overlapping-source scenarios, highlighting the necessity of explicit 3D modeling for robust spatial reasoning in complex environments.

### 5.3. Ablation Studies

We conduct a comprehensive ablation studies to validate our core design choices, focusing on efficacy of learnable spatial audio representations, depth-aware visual grounding and the both individual contributions to joint audio-visual reasoning tasks.

**Effectiveness and Generalization of Neural IV.** We compare the proposed Neural IV against the Classical IV to evaluate robustness in complex acoustic environments. As shown in Table 3, Neural IV consistently achieves lower angular errors across all settings. To test generalization, we perform a cross-evaluation where models are trained on one source type (*e.g.*, Single Source) and tested on the other (*e.g.*, Overlap Source). The results indicate that while Classical IV suffers significant degradation in mismatched scenarios, Neural IV exhibits superior stability. This suggests that Neural IV learns robust, intrinsic spatial acoustic.

**Impact of Depth Encoding on Visual Grounding.** We investigate the contribution of our 3D-aware positional encoding to the Visual Grounding task (Task C). Table 4 demonstrates that explicitly integrating depth information is critical for precise localization. The inclusion of depth encoding improves the Mean 3D IoU from 0.29 to 0.32 and reduces the Median Visual Offset from 0.18m to 0.16m. These improvements confirm that explicit geometric priors enable the model to better map 2D visual features to metric 3D

*Table 4.* Ablation study on **Depth Encoding** for visual grounding. We report both Mean and Median values to better analyze the distribution of errors.

| Model Setting | 3D IoU ↑ | | Visual Offset ↓ | |
|---|---|---|---|---|
| | Mean | Median | Mean | Median |
| Full Model (w/ Depth) | **0.32** | **0.31** | **0.22** | **0.16** |
| w/o Depth Encoding | 0.29 | 0.27 | 0.24 | 0.18 |

*Table 5.* Component analysis on **Reasoning Tasks**. We evaluate the impact of audio encoders (Neural/Classical), Depth, and FOA on 1-speaker and 2-speaker scenarios.

| Model Setting | Accuracy (%) | |
|---|---|---|
| | 1-Speaker ↑ | 2-Speaker ↑ |
| Ours (Neural IV) | **99.5** | **99.2** |
| Ours (Classical IV) | 99.5 | 98.6 |
| Ours (Neural) w/o Depth | 96.9 | 94.9 |
| Ours (Classical) w/o Depth | 99.2 | 98.7 |
| Ours w/o FOA Encoder | 43.8 | 47.6 |
| Ours w/o Depth & FOA | 43.8 | 45.7 |

coordinates, reducing the ambiguity inherent in monocular RGB perception.

**Component Analysis on Joint Reasoning.** Finally, we dissect the contribution of each core component to the joint audio-visual reasoning tasks (Tasks D & E) in Table 5. First, Neural IV consistently yields higher accuracy than Classical IV, particularly in the challenging 2-speaker scenario (99.2% vs. 98.6%). Second, removing depth encoding leads to a noticeable drop in accuracy, validating that depth perception assists in spatially distinguishing potential target speakers. Most critically, removing the FOA encoder causes reasoning performance to collapse to near-random levels (∼43–47%), even after fine-tuning. This catastrophic drop empirically demonstrates that standard monaural representations are fundamentally insufficient for spatial disambiguation, underscoring the indispensable role of our native 3D architecture in solving complex physical reasoning tasks.

## 6. Conclusion

In this paper, we introduced **JAEGER**, a novel end-to-end framework that bridges the dimensionality gap between current 2D-centric AV-LLMs and a simulated 3D physical environment. By synergizing RGB-D visual perception with multi-channel FOA, our model achieves precise joint localization and reasoning capabilities previously unattainable by monaural, RGB-only systems. We further proposed Neural IV, a data-driven acoustic representation that significantly

enhances robustness in challenging, reverberant scenarios. We constructed SPATIALSCENEQA, the first large-scale, high-fidelity dataset for 3D audio-visual instruction tuning. Our extensive experiments show that explicit 3D modeling of both visual depth and spatial audio is indispensable for solving complex physical reasoning tasks. We hope this work advances the development of embodied agents with holistic 3D perception and interaction capabilities.

## Impact Statement

This paper aims to advance machine learning systems that can perceive and reason about 3D physical environments using synchronized visual geometry and spatial audio. Such capabilities may benefit embodied AI, assistive perception, and human–machine interaction in spatially complex environments. However, our current evaluation is primarily simulation-based, and performance may not transfer cleanly to real acoustics, microphones, RGB-D sensors, synchronization pipelines, or calibration settings without additional validation. Stronger audio-visual spatial grounding may also raise privacy risks if used for invasive localization or surveillance. We therefore recommend that any real-world deployment require informed consent, data minimization, limited retention, access control, and careful documentation of sensor and domain limitations.

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

## A. Real-World Transfer on STARSS23

To assess whether the spatial audio representations learned from SpatialSceneQA transfer beyond simulation, we conduct a small-scale experiment on STARSS23 (Shimada et al., 2023), a real-world first-order ambisonics (FOA) benchmark. We construct a subset consisting only of single-speaker clips with approximately stationary sources, so that the setting better matches our source-localization evaluation protocol. The resulting real-data subset is limited, containing 753 clips and 3.84 hours of audio, so this experiment should be interpreted as a sanity check rather than a complete real-world validation. Nevertheless, initialization from SpatialSceneQA consistently improves over training from scratch, suggesting that the learned FOA features capture transferable spatial cues instead of merely overfitting to the simulator.

As shown in Table 6, SpatialSceneQA pretraining improves both Neural IV and Classical IV on real FOA localization. The improvement is especially clear for elevation, where Neural IV reduces the median error from $7.3°$ to $4.8°$. Azimuth remains more challenging under real acoustics, but pretraining still reduces the median azimuth error for both variants. These results support our main claim that explicit 3D audio-visual modeling is learnable in a controlled simulation setting and provides encouraging transfer to real FOA localization, while not implying that sim-to-real transfer is fully solved.

*Table 6.* Small-scale STARSS23 real-world FOA localization results. We compare models initialized from SpatialSceneQA pretraining against models trained from scratch. Lower median angular error is better.

| Model | Elevation Median Error (°) ↓ | | Azimuth Median Error (°) ↓ | |
|---|---|---|---|---|
| | Pretrained | Scratch | Pretrained | Scratch |
| Neural IV | 4.8 | 7.3 | 76.7 | 94.9 |
| Classical IV | 7.0 | 8.0 | 74.2 | 99.7 |

## B. Matched-Input Architectural Ablation

To isolate the contribution of the proposed architectural components from the benefit of using richer modalities, we add a matched-input ablation that keeps the same RGB-D and FOA inputs but replaces JAEGER with a much simpler fusion strategy. For RGB-D, the depth map is copied to three channels and concatenated with RGB along the image-width dimension. For FOA, the four channels are flattened into one long waveform and processed as if it were monaural audio. This *SimpleFuse* baseline therefore sees the same input modalities but removes the depth-aware visual encoding and structured FOA spatial modeling.

Table 7 shows that SimpleFuse is consistently worse than both JAEGER variants across audio localization, visual grounding, and joint reasoning. The gap is most pronounced in the audio-visual reasoning tasks, where SimpleFuse obtains $67.79\%$ and $46.59\%$ accuracy, compared with $99.5\%$ / $98.6\%$ for JAEGER with Classical IV and $99.5\%$ / $99.2\%$ for JAEGER with Neural IV. This indicates that the gains do not come merely from exposing the model to RGB-D and FOA inputs, but from how JAEGER represents and aligns 3D geometry and spatial audio.

*Table 7.* Matched-input architectural ablation. A-O-1 and A-O-2 denote single-source and overlapping-source audio-only DoA median error; V-O IoU and V-O Offset denote visual-only 3D grounding IoU and center offset; A-V-1 and A-V-2 denote single-speaker and two-speaker audio-visual reasoning accuracy.

| Model | A-O-1 (°) ↓ | A-O-2 (°) ↓ | V-O IoU ↑ | V-O Offset (m) ↓ | A-V-1 (%) ↑ | A-V-2 (%) ↑ |
|---|---|---|---|---|---|---|
| SimpleFuse | 3.40 | 21.90 | 0.30 | 0.17 | 67.79 | 46.59 |
| JAEGER (Classical IV) | 2.95 | 6.44 | 0.32 | 0.16 | 99.5 | 98.6 |
| JAEGER (Neural IV) | 2.21 | 4.11 | 0.32 | 0.16 | 99.5 | 99.2 |

## C. Comparison between FOA and Binaural Spatial Audio

We further compare FOA against binaural spatial audio under a matched JAEGER-style setup. We keep the same Qwen audio encoder used in the FOA model, replace the FOA spatial branch with binaural embeddings extracted by the Spatial-AST encoder of BAT (Zheng et al., 2024), and concatenate these binaural embeddings with the same Qwen audio embeddings. This gives a binaural variant, denoted JAEGER (BAT), whose pipeline is aligned with the FOA variants except for the spatial-audio representation.

Table 8 shows that FOA-based JAEGER remains stronger than the binaural variant on the key tasks in our benchmark. The gap is clearest on audio-visual reasoning, especially in the harder two-speaker setting, where JAEGER (BAT) reaches 85.0% accuracy while JAEGER (Classical IV) and JAEGER (Neural IV) reach 98.6% and 99.2%, respectively. This result does not imply that FOA is universally superior to binaural audio, but it supports that, in our matched setting, FOA provides a more effective spatial representation for the studied 3D audio-visual grounding and reasoning tasks.

*Table 8.* FOA versus binaural spatial audio under a matched JAEGER-style setup. Audio DoA and Overlap DoA report median angular error; 1-speaker and 2-speaker report audio-visual reasoning accuracy.

| Model | Modality | Audio DoA ↓ | Overlap DoA ↓ | 1-speaker ↑ | 2-speaker ↑ |
|---|---|---|---|---|---|
| JAEGER (BAT) | RGB-D + Binaural | 7.73° | 8.25° | 95.9 | 85.0 |
| JAEGER (Classical IV) | RGB-D + FOA | 2.95° | 6.44° | 99.5 | 98.6 |
| JAEGER (Neural IV) | RGB-D + FOA | 2.21° | 4.11° | 99.5 | 99.2 |

## D. Harder Multi-Candidate Reasoning Setting

The original audio-visual reasoning benchmark uses two to three visible candidate speakers. To better calibrate the near-ceiling reasoning accuracy, we add a harder setting by increasing the number of visible candidates to four to six. The harder-split models are trained with only a mini training set of 160 training samples per candidate-number setting, which is substantially smaller than the original training scale of at least 4k samples. We therefore view this experiment primarily as a test of generalization to harder candidate-cardinality settings rather than as a fully optimized upper bound.

As shown in Table 9, accuracy decreases as the distractor set grows. In the single-source reasoning setting, JAEGER (Classical IV) drops from 87.5% with four candidates to 80.0% with six candidates, while JAEGER (Neural IV) drops from 95.0% to 72.5%. This confirms that the original near-ceiling results should be interpreted in the context of a controlled benchmark, and that increasing the number of plausible visual distractors makes the reasoning task substantially more challenging.

*Table 9.* Single-source audio-visual reasoning accuracy under harder settings with more visible speaker candidates.

| Model | 4 Candidates ↑ | 5 Candidates ↑ | 6 Candidates ↑ |
|---|---|---|---|
| JAEGER (Classical IV) | 87.5 | 82.5 | 80.0 |
| JAEGER (Neural IV) | 95.0 | 77.5 | 72.5 |

## E. Coordinate System Definition

In this work, we adopt a right-handed Cartesian coordinate system defined as follows: the $x$-axis points to the right, the $y$-axis points vertically upwards, and the $z$-axis points backwards.

Based on this configuration, the spherical coordinates (azimuth and elevation) are defined relative to the frontal view:

- **Azimuth:** We define $0°$ as the forward direction, aligned with the negative $z$-axis. Positive values indicate a rotation to the left. The range of azimuth is $[-180°, 180°]$.

- **Elevation:** We define $0°$ as the horizontal plane. Positive values indicate an upward direction. The range of elevation is $[-90°, 90°]$.

## F. Diversity Analysis of Speaker Point Clouds

In this section, we assess the diversity of the generated 3D assets. We utilized the Hunyuan3D-1 text-to-point-cloud generation framework to synthesize a total of 120 unique speaker instances, which serve as visual cues for sound sources in our experiments. The dataset was partitioned into training, validation, and testing sets following an 8:1:1 ratio.

All models were generated using the consistent text prompt "floor standing speaker." Crucially, to ensure sufficient morphological diversity within the dataset, we iteratively updated the random seed for each generation instance. Figure 4 visualizes a random subset of 32 generated point clouds, providing a qualitative demonstration of the structural variety achieved in our dataset.

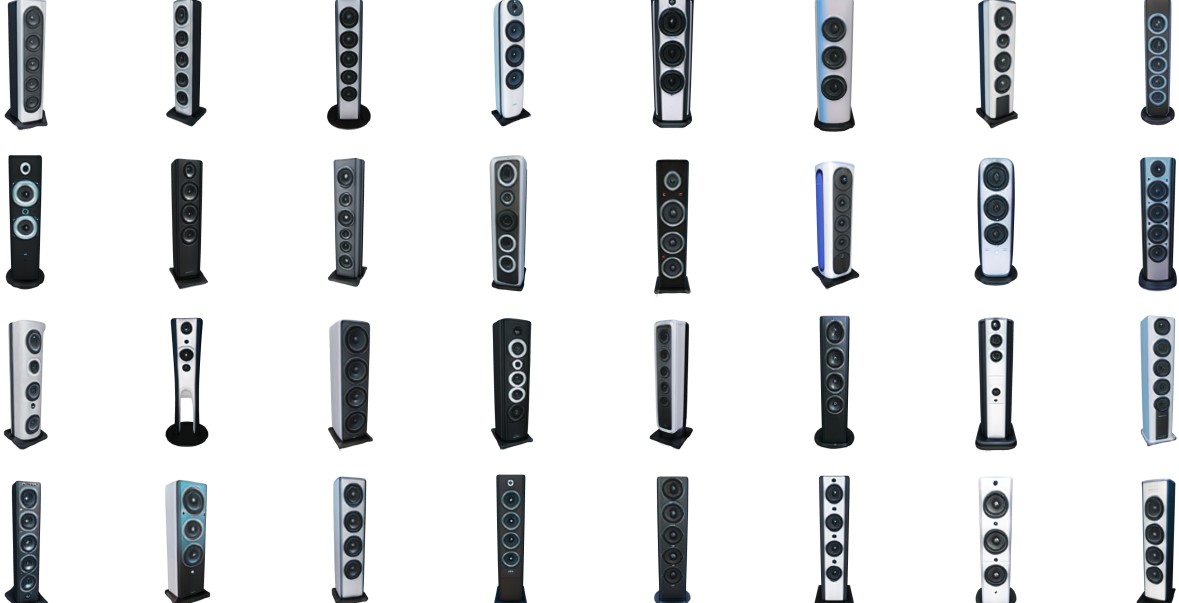

*Figure 4.* Visualization of the diversity in generated speaker point clouds. We display 32 randomly selected samples from the 120 generated instances. Despite using the same text prompt, varying the random seed results in distinct structural and morphological variations.

