# OpenReview forum: "JAEGER: Joint 3D Audio-Visual Grounding and Reasoning in Simulated Physical Environments"
_ICML.cc/2026/Conference — ICML 2026 regular_

### Official Review · Reviewer_GNaY · 2026-03-02

**Soundness:** 2
**Presentation:** 3
**Significance:** 2
**Originality:** 2
**Overall Recommendation:** 4
**Confidence:** 4

**Summary:**

This paper introduces JAEGER, an end-to-end framework that extends current audio-visual LLMs from 2D to 3D spatial grounding and reasoning by jointly modeling RGB-D observations and 4-channel first-order ambisonics audio. The method disentangles audio semantics from spatial cues and proposes a learnable Neural Intensity Vector to replace STFT-based intensity features for more robust direction-of-arrival (DoA) estimation under reverberation and overlapping sources. To train and evaluate the model at scale, the authors build SpatialSceneQA, synthesized in simulated environments with calibrated geometry and dense 3D annotations spanning DoA estimation, 3D box grounding, and multi-speaker audio-visual matching. Experiments show improved DoA errors with Neural IV and strong performance on joint reasoning tasks in simulation, suggesting explicit 3D modeling of both vision and audio is beneficial for physical-environment reasoning.

**Compliance With Llm Reviewing Policy:**

Affirmed.

**Final Justification:**

Thanks to the authors for the rebuttal and clarification. I do not consider this a particularly strong paper, but their responses have partially alleviated my concerns, so I raise my score to 4.

**Key Questions For Authors:**

1. Do you have any real-world sanity-check results (even small-scale) using a commodity RGB-D sensor (e.g., RealSense) together with a real FOA/ambisonics recording for DoA and/or matching, and if not, what do you expect the main sim-to-real failure modes to be?

2. Your data generation restricts cases to same-room source–receiver pairs with 1–4m distance and a visibility threshold. How sensitive are the results if you relax these constraints and evaluate performance stratified by distance and occlusion?

3. In the matching tasks, candidates are typically 2–3 loudspeakers, and accuracy is near-saturated. Have you tested larger candidate sets (e.g., 5/8/10) or harder distractors, and how does performance scale with the number of candidates

4. Since there is no baseline with exactly the same RGB-D+FOA inputs, can you provide a matched-input baseline (same inputs, but using a simpler fusion and/or traditional STFT-based IV instead of Neural IV) to isolate what portion of the gain comes from the method design rather than richer modalities?

5. For the multi-speaker setting, prompts include gender labels to specify the target. Have you ablated this by removing/perturbing identity cues and using only spatial descriptions (left/right/near/far), to verify the model is not relying on a textual shortcut?

**Limitations:**

No. The paper would benefit from a short limitations/impact section that (i) notes the current evaluation is simulation-based and may not transfer cleanly to real acoustics/sensors, and (ii) discusses potential misuse such as privacy-invasive localization or surveillance enabled by stronger audio-visual spatial grounding, along with practical mitigations.

**Strengths And Weaknesses:**

**Strengths**:

1. The paper clearly formulates joint 3D audio-visual grounding by combining RGB-D inputs with FOA spatial audio in a single end-to-end setup.

2. The proposed Neural Intensity Vector is a straightforward, learnable replacement for hand-crafted intensity features to encode FOA directional cues for DoA-related reasoning.

3. The SpatialSceneQA benchmark spans multiple spatial tasks with aligned RGB-D and FOA supervision, enabling broader evaluation than a single-task setup.


**Weakness**:

1. SpatialSceneQA is synthesized using SoundSpaces 2.0 in scanned 3D environments (HM3D) with rendered RGB-D and FOA audio, which is valuable for control but does not capture real sensor noise and recording artifacts. As a result, it is unclear how the learned FOA/RGB-D fusion (and Neural IV) will behave on real RGB-D cameras and real ambisonics recordings.

2. The pipeline samples source–receiver pairs within the same room and restricts distance to 1–4m, and it enforces a visibility constraint that removes heavily occluded or distant targets. These constraints reduce ambiguity (e.g., long-range attenuation, severe occlusion, and complex multi-room propagation), so the reported results may not reflect performance in harder, more realistic layouts.

3. The joint reasoning accuracy is reported as 99.2%, while the matching tasks are small-choice identification among 2–3 visible loudspeakers (and the dataset construction guarantees at least two visible candidates). With this level of saturation and limited candidate set size, it is hard to diagnose what failure modes remain or whether the model is robust under more distractors and stronger confounders.

4. The paper notes Qwen2.5-Omni is monaural-only and thus inherently incapable of DoA, while other baselines are either specialized single-modality models or random guessing. Without a strong baseline that uses the same RGB-D+FOA inputs, it is difficult to isolate gains from the specific architectural choices (e.g., Neural IV, fusion design) versus simply providing richer modalities.

5. Audio sources are dry monaural speech from LibriSpeech, and the reasoning prompts even use provided gender labels to specify the target speaker, and the visual sound-emitting objects are primarily inserted into loudspeaker meshes (with visibility filtering). This design makes the task well-controlled, but it leaves open whether the method works for non-speech sounds, more diverse emitting objects, or cases where identity cues are not available in the instruction.

---

> ### Author Rebuttal · Authors · 2026-03-31
>
> We thank the reviewer for the careful reading and constructive suggestions. We address the main points below.
>
>  - **W1 / Q1: Real-world transfer and sanity check.**
>
> We do not currently have a commodity RGB-D sensor+real FOA/ambisonics testbed (e.g., RealSense + FOA microphone) for full joint RGB-D+FOA evaluation. To assess transfer beyond simulation, we conducted a small-scale experiment on STARSS23 [1], a real-world FOA benchmark. Although real data are limited (753 clips, 3.84 hours), initialization from SpatialSceneQA consistently outperforms training from scratch, suggesting that the learned FOA representation is transferable rather than merely overfitting to simulation. In particular, elevation localization also remains strong on real data. Expected sim-to-real failure modes include microphone-array mismatch, acoustic-material mismatch, limitations of the spatial-audio simulation itself, and synchronization/calibration errors. Detailed results are in our response to **Reviewer dJWu, W1 / Q1**. We will make these limitations clearer in the paper.
>  - **W2 / Q2: Sensitivity to distance / occlusion.**
>
> We added a distance-stratified analysis on the harder multi-candidate setting. Since visible target size in the image generally decreases as distance increases, this also partially reflects the effect of visibility/occlusion difficulty. The tables below report accuracy (%) on the harder single-source AV reasoning task with enlarged candidate sets. Some cells reach 100% accuracy mainly because the test set is small. For 4–6 visible speakers, both JAEGER variants remain reasonably stable across 1–4 m (distances beyond 5 m are difficult to sample at scale in HM3D because of room-size limitations), without a monotonic collapse.
> |Distance|Accuracy of JAEGER-Classical (4spk)(%)↑|Accuracy of JAEGER-Classical (5spk)(%)↑|Accuracy of JAEGER-Classical (6spk)(%)↑|
> |:-:|:-:|:-:|:-:|
> |1–2m|89.66|82.61|76.67|
> |2–3m|87.50|81.82|100.00|
> |3–4m|66.67|83.33|75.00|
>
> |Distance|Accuracy of JAEGER-Neural (4spk)(%)↑| Accuracy of JAEGER-Neural (5spk)(%)↑|Accuracy of JAEGER-Neural (6spk)(%)↑|
> |:-:|:-:|:-:|:-:|
> |1–2m|93.1|78.26|66.67|
> |2–3m|100.00|72.73|83.33|
> |3–4m|100.00|83.33|100.00|
>  - **W3 / Q3: More distractors / harder reasoning settings.**
>
> To better calibrate the original near-ceiling reasoning accuracy, we added a harder setting by increasing the number of visible candidate speakers from the original 2–3 to 4–6. Under this split, performance already decreases as the distractor set grows. Notably, these harder-split models were trained with only a mini training set (160 training samples per candidate-number setting), much smaller than the original training scale (at least 4k samples), so we view this mainly as a test of generalization to harder candidate-cardinality settings. For the detailed results, please refer to our response to **Reviewer x7KA, W3 / Q3**.
>  - **W4 / Q4: Matched-input baseline to isolate architectural gains.**
>
> We added a matched-modality ablation that keeps the same RGB-D + FOA inputs but replaces our design with a much simpler fusion strategy. For RGB-D, the depth map is copied to 3 channels and concatenated with RGB along the width dimension; for FOA, the 4 channels are flattened into one long waveform and treated as monaural audio. This simple matched-input fusion is consistently much worse than both JAEGER variants, indicating that the gains do not come merely from richer modalities, but from how JAEGER represents and aligns them. For the detailed results, please refer to our response to **Reviewer dJWu, W3**.
>  - **W5 / Q5: Non-speech sounds, more diverse emitters, and identity cues.**
>     - We agree that extending beyond speech and loudspeaker-style emitters would be valuable; we view this as an important direction for future work.
>     - For the current overlapping-speaker reasoning task, the identity cue (e.g., gender) is used to specify the target audio stream among overlapping sources. Our intended reasoning chain is: first identify the target source in the overlapping audio via semantic cue, then localize that source spatially, then match it to the visually localized candidate speakers. Replacing it with purely spatial descriptions such as left/right/near/far would instead introduce a shortcut, because the model could often answer from the visual layout alone, without actually using the mixed audio to identify which source is being queried. We will clarify this task design more explicitly in the revision.
>  - **Limitations.**
>
> We will add a limitation/impact section noting that our evaluation is simulation-based and may not fully transfer to real acoustics/sensors, and that stronger audio-visual spatial grounding may raise privacy risks. We will also discuss mitigations including consent, limited retention, and access control.
>
> [1] Shimada K, et al. STARSS23: An Audio-Visual Dataset of Spatial Recordings of Real Scenes with Spatiotemporal Annotations of Sound Events. In Proc. NeurIPS, 2023.

---

> > ### Author Rebuttal · Reviewer_GNaY · 2026-04-01
> >
> > Thank you for the detailed rebuttal and the additional analyses. Some of my concerns are partially addressed. However, my concern about the lack of real RGB-D + FOA validation is not fully resolved.

---

> > > ### Author Response · Authors · 2026-04-08
> > >
> > > We agree that the lack of real RGB-D + FOA validation is not fully resolved and remains a limitation of the current work. To probe this issue a bit more directly, beyond the audio-only STARSS23 localization experiment discussed in our earlier rebuttal, we also conducted a separate small-scale audio-visual pilot on STARSS23. Specifically, we constructed a two-speaker subset with visible speakers and asked the model to match a queried speech segment to the corresponding visible speaker (a left/right decision). On this pilot, the model achieved 56.47% accuracy (131/232).
> > >
> > > We stress, however, that we do not view this result as conclusive evidence of real-world joint RGB-D + FOA capability. The derived AV subset is extremely small (24 min training / 16 min test) and has limited scene diversity, with highly similar environments and a narrow distribution of speaker/layout configurations. Moreover, STARSS23 does not provide depth, so the depth input in this pilot was estimated using MoGe-2 [1], rather than captured by a real RGB-D sensor. Under these conditions, the experiment is substantially underpowered for evaluating generalization of joint RGB-D + FOA reasoning and is also too limited for reliably attributing the gap to any specific sim-to-real failure mode.
> > >
> > > That said, since we still do not have a full commodity RGB-D + real FOA testbed, we expect the main sim-to-real failure modes to remain microphone-array mismatch, acoustic-material and room-response mismatch, synchronization/calibration errors between modalities, and geometry/depth mismatch. We therefore view this pilot primarily as a limited sanity check, rather than a decisive validation of real RGB-D + FOA performance.
> > >
> > > [1] Wang R, et al. MoGe-2: Accurate Monocular Geometry with Metric Scale and Sharp Details. In Proc. NeurIPS, 2025.

---

### Official Review · Reviewer_x7KA · 2026-03-12

**Soundness:** 3
**Presentation:** 3
**Significance:** 3
**Originality:** 3
**Overall Recommendation:** 4
**Confidence:** 3

**Summary:**

This paper studies joint 3D audio-visual grounding and reasoning in simulated physical environments. The proposed method, JAEGER, extends a recent audio-visual multimodal LLM with RGB-D input, depth-aware visual encoding, and FOA-based spatial audio. The paper also introduces Neural IV, a learned spatial audio representation, and SpatialSceneQA, a synthetic benchmark covering localization, 3D grounding, and multi-speaker matching. The experiments suggest that explicit depth and spatial audio are useful for the proposed tasks, especially in more challenging localization and reasoning settings.

**Compliance With Llm Reviewing Policy:**

Affirmed.

**Key Questions For Authors:**

1. **Table 5 clarification:** What exactly is retained in the “w/o FOA encoder” setting? Does this ablation keep the monaural or omnidirectional audio pathway and remove only the spatial FOA branch, or does it remove audio conditioning more broadly? This affects how I interpret the component analysis.
2. **FOA vs. spatial audio more generally:** Since the paper already converts FOA to binaural audio for BAT, could the authors include a binaural reasoning ablation, or at least discuss how they expect such a comparison to behave? This would clarify whether the gain comes from FOA specifically or from spatial audio more generally.
3. **Reasoning difficulty:** Can the authors report harder splits or additional breakdowns for the reasoning tasks, for example with more distractors, more occlusion, or more difficult geometry? This would help calibrate the meaning of the very high reasoning accuracy.

**Limitations:**

Yes

**Strengths And Weaknesses:**

## Strengths

**S1. The paper addresses a relevant problem.**

The motivation is clear: many recent audio-visual multimodal LLMs still rely mainly on RGB video and monaural audio, which makes precise spatial grounding difficult. The paper identifies this limitation well and focuses on a problem that is worth studying.

**S2. The method is coherent and well matched to the task.**

The overall design is sensible. The visual stream explicitly uses depth information, and the audio stream separates semantic content from spatial cues. Neural IV also fits naturally into the paper’s goal of improving spatial audio understanding under more difficult conditions.

**S3. The dataset contribution is meaningful.**

SpatialSceneQA is a substantial part of the paper’s contribution. It provides synchronized RGB-D, FOA audio, and 3D annotations, and it covers several related tasks rather than only one narrow benchmark. This makes the work more useful than a model-only contribution.

**S4. The experiments support several of the main claims.**

The main results and ablations are generally aligned with the paper’s arguments. In particular, the comparisons between Classical IV and Neural IV, the depth ablation, and the component analysis for reasoning all help clarify what each part of the system is doing.

## Weaknesses

**W1. The comparison set is still somewhat heterogeneous.**

The paper is transparent that there is no direct baseline with the same modality setup, but the resulting comparisons mix fairly different kinds of systems: an audio-only spatial model, zero-shot visual grounding models, and a monaural omni model. These results are still informative, but they make the empirical picture less clean than ideal.

**W2. The Table 5 FOA ablation does not fully isolate the benefit of FOA itself.**

The current component analysis mainly shows that monaural audio is insufficient for the reasoning task. That is useful, but it is not the same as showing that FOA is specifically better than other spatial audio formats. Since the paper already converts FOA to binaural for BAT in the main comparison, a binaural reasoning ablation would have made this point much stronger.

**W3. The reasoning benchmark seems fairly controlled.**

The data construction uses same-room source–receiver pairs, limited source distance, visibility filtering, and only a small number of candidate speakers in the reasoning tasks. These choices are reasonable, but they also make the near-ceiling reasoning results a bit harder to interpret. More evidence on harder settings would help.

---

> ### Author Rebuttal · Authors · 2026-03-31
>
> We thank the reviewer for the positive assessment and constructive suggestions. We respond to the main points below.
>  - **W1: The comparison set is somewhat heterogeneous.**
>
> We agree that the comparison table is somewhat heterogeneous and not as clean as an ideal matched-modality benchmark. This mainly reflects the current state of the field: there is still no direct baseline with the same RGB-D + FOA setup and the same joint 3D audio-visual grounding/reasoning tasks. To more clearly situate our model relative to the strongest available methods for the different subproblems, we therefore placed these neighboring baselines in a single table, including audio-only spatial localization, visual grounding, and monaural omni models. We will make this motivation clearer in the revision.
>
>  - **W2 / Q2: FOA vs. binaural more generally.**
>
> We appreciate this suggestion that a direct FOA-vs-binaural comparison for the reasoning task would make the scope of the claim clearer. Our intention in this paper is not to claim that FOA is universally better than binaural. Rather, the main point is that the gain comes from explicit spatial audio more generally, together with explicit 3D geometry, rather than from FOA alone. In terms of expected behavior, we would expect FOA and binaural to be broadly comparable in this setting, since both provide spatial information and should therefore both be more informative than monaural audio for the reasoning task. A more systematic comparison between FOA and other spatial audio formats, including binaural audio, would be a valuable extension, and we view it as an important direction for future work.
>
>  - **W3 / Q3: The reasoning benchmark seems fairly controlled; harder settings would help.**
>
> We appreciate this suggestion. To better calibrate the original reasoning results, we added a harder setting by increasing the number of visible candidate speakers from the original 2–3 to 4–6. Under this harder split, performance decreases substantially as the number of distractors grows, which helps contextualize the original near-ceiling accuracy. In the single-source reasoning setting, JAEGER-Classical drops from 87.5% at 4 candidates to 80.0% at 6 candidates, while JAEGER-Neural drops from 95.0% to 72.5%. Notably, these harder-split models were trained with only a mini training set (160 training samples per candidate-number setting), which is much smaller than the original training scale (at least 4k samples). We therefore view this experiment primarily as a test of generalization to harder candidate-cardinality settings, rather than as a fully optimized upper bound. Even under this lightweight setting, the clear performance drop shows that the reasoning task becomes significantly more challenging once the distractor set is enlarged. We will include this harder-split evaluation to better interpret the original reasoning scores.
>
> | # Visible speaker candidates | 4 | 5 | 6 |
> |:---:|:---:|:---:|:---:|
> | Accuracy of JAEGER-Classical, A-V reasoning (1-source) (%) ↑ | 87.5 | 82.5 | 80.0 |
> | Accuracy of JAEGER-Neural, A-V reasoning (1-source) (%) ↑ | 95.0 | 77.5 | 72.5 |
>
>  - **Q1: What exactly is retained in the “w/o FOA encoder” setting?**
>
> In the “w/o FOA encoder” setting, we remove only the FOA IV encoder, while retaining the omnidirectional audio pathway for semantic audio content. Thus, this ablation is intended to test whether monaural/omnidirectional audio semantics alone are sufficient for the reasoning task.

---

> > ### Author Rebuttal · Reviewer_x7KA · 2026-04-03
> >
> > The rebuttal meaningfully addresses my questions about the interpretation of the FOA ablation and the ease of the original reasoning benchmark. In particular, the clarification that “w/o FOA encoder” retains the omnidirectional semantic audio pathway, together with the newly added harder 4–6 candidate split and matched-modality SimpleFuse ablation, strengthens the empirical story. However, my concern that the current evidence does not isolate the benefit of FOA specifically relative to other spatial audio formats (e.g., binaural) remains. Therefore, I view the rebuttal as partially resolving my concerns; it supports maintaining my current evaluation, but not raising it.

---

> > > ### Author Response · Authors · 2026-04-08
> > >
> > > Thank you for clarifying that the remaining concern is whether the current evidence isolates the benefit of FOA specifically relative to other spatial audio formats, such as binaural. To address this point directly, we added a new FOA-vs-binaural comparison under a matched setup. Specifically, we kept the same Qwen audio encoder as in our FOA model, replaced the FOA input branch with binaural embeddings extracted by the Spatial-AST encoder of BAT [1], and concatenated them with the same Qwen audio embeddings so that the overall pipeline remains aligned with the FOA setting. We then compared this binaural variant, denoted as JAEGER (BAT), against our original JAEGER (Classical IV) and JAEGER (Neural IV) models.
> > >
> > > The resulting comparison is summarized below. As shown, FOA-based JAEGER remains stronger than the binaural variant on the key tasks of this work, with the clearest gap appearing on the audio-visual reasoning benchmarks, especially the harder 2-speaker setting. These additional results support that the improvement is not only from adding spatial audio in a generic sense; in our matched setting, FOA provides a more effective representation than binaural for the 3D audio-visual grounding and reasoning tasks studied in this paper.
> > > | Model                 | Modality         | Audio DoA ↓ | Overlap DoA ↓ | 1-speaker ↑ | 2-speaker ↑ |
> > > | :-------------------- | :--------------- | :---------: | :-----------: | :---------: | :---------: |
> > > | JAEGER (BAT)          | RGB-D + Binaural |    7.73°    |    18.21°    |     95.9    |     85.0    |
> > > | JAEGER (Classical IV) | RGB-D + FOA      |    2.95°    |     16.09°    |     99.5    |     98.6    |
> > > | JAEGER (Neural IV)    | RGB-D + FOA      |    2.21°    |     13.13°    |     99.5    |     99.2    |
> > >
> > >
> > > [1] Zheng Z, et al. BAT: Learning to Reason about Spatial Sounds with Large Language Models. In Proc. ICML, 2024.

---

### Official Review · Reviewer_dJWu · 2026-03-13

**Soundness:** 3
**Presentation:** 3
**Significance:** 3
**Originality:** 2
**Overall Recommendation:** 4
**Confidence:** 2

**Summary:**

Thw paper proposes JAEGER, an end-to-end framework for extending audio-visual LLMs from 2D RGB plus monaural audio to 3D reasoning with RGB-D input and 4-channel first-order ambisonics. The method combines depth-aware visual encoding, a dual-path audio design, and a learned spatial audio representation called Neural IV, and is trained on a newly constructed synthetic benchmark, SpatialSceneQA, containing 61k instruction-tuning samples. The paper evaluates audio direction-of-arrival estimation, 3D visual grounding of sound-emitting objects, and multi-speaker audio-visual matching, and reports strong gains over 2D-centric baselines and over a classical intensity-vector variant.

**Compliance With Llm Reviewing Policy:**

Affirmed.

**Final Justification:**

Firstly I would like to clarify I am not an expert at this topic and hence would rely on the consensus of other reviewers and the AC. Right now it seems the consensus is diverging. I am deciding to raise my score from 3 to 4 as the rebuttal seems reasonable to me. However, I would leave the final decision to the other reviewers and AC who are relatively more knowledgable on the topic.

**Key Questions For Authors:**

1. Can the method transfer well for scenes beyond SpatialSceneQA?

**Limitations:**

yes

**Strengths And Weaknesses:**

**Strengths**
1. The paper addresses a meaningful problem of adapting audio-visual LLMs for 3D spatial grounding.

**Weaknesses**

1. The evaluations are done only on synthetic benchmarks. This raises concern whether the framework can be applicable in the real world.
2. The paper should compare runtime. computational cost and latency of the proposed framework as compared to the existing baselines.
3. More ablation studies are required to isolate the gains coming from each architectural component of the proposed framework.
4. The visual grounding results are not very impressive considering IoU of 0.32 on a synthetic benchmark dataset.

---

> ### Author Rebuttal · Authors · 2026-03-31
>
> We thank the reviewer for the constructive feedback. We respond to each concern below.
> - **W1/Q1: Synthetic-only evaluation and transfer beyond SpatialSceneQA.**
>
> We agree that real-world transfer is important. Our goal is to establish a controlled benchmark and end-to-end framework for jointly studying RGB-D geometry and FOA spatial audio, since large-scale real data with synchronized RGB-D, ambisonics, calibration, source poses, and 3D annotations is difficult to obtain. To assess transfer beyond simulation, we additionally conducted a small-scale experiment on STARSS23 [1], a real-world FOA benchmark. Although the real-data scale is limited (753 clips, 3.84 hours), initialization from SpatialSceneQA consistently outperforms training from scratch, suggesting that the learned FOA representation captures transferable cues rather than only overfitting to simulation. In particular, our model localizes elevation well in real scenes. We will include these results and clarify that our claim is not that sim-to-real transfer is solved, but that explicit 3D audio-visual modeling is learnable in a controlled setting and shows encouraging transfer to real FOA localization.
> |STARSS23|Elevation median error(°)↓ Pretrained|Elevation median error(°)↓ Scratch|Azimuth median error(°)↓ Pretrained|Azimuth median error(°)↓ Scratch|
> |:-:|:-:|:-:|:-:|:-:|
> |NeuralIV|4.8|7.3|76.7|94.9|
> |ClassicalIV|7.0|8.0|74.2|99.7|
> - **W2: The paper should compare runtime, computational cost and latency with existing baselines.**
>
> We profiled Qwen2.5-Omni, JAEGER-Classical, and JAEGER-Neural under the same benchmark pipeline. Qwen2.5-Omni has 8.93B total parameters, while JAEGER-Classical and JAEGER-Neural have 9.09B and 9.10B. In terms of inference memory, Qwen2.5-Omni uses 17.31 GB peak GPU memory, compared with 17.95 GB for JAEGER-Classical and 18.04 GB for JAEGER-Neural, indicating only modest overhead. We also report observed end-to-end latency: 0.340 s for JAEGER-Classical, 0.369 s for JAEGER-Neural, and 1.928 s for Qwen2.5-Omni. These values should be interpreted with care: JAEGER is task-finetuned to output short structured answers directly (2 tokens on average), whereas Qwen2.5-Omni is not task-finetuned and tends to generate much longer free-form responses (about 51 tokens). Thus, these numbers are best viewed as observed runtime under this benchmark setting, rather than a strict decoding-efficiency comparison.
> |Model|Total params|Peak inference GPU memory (GB)|End-to-end latency (s)|
> |:-:|:-:|:-:|:-:|
> |Qwen2.5-Omni|8.93B|17.31|1.928|
> |JAEGER-Classical|9.09B|17.95|0.340|
> |JAEGER-Neural|9.10B|18.04|0.369|
> - **W3: More ablation studies are required to isolate the gains from each architectural component.**
>
> We added a matched-modality ablation that keeps the same RGB-D + FOA inputs but replaces our method with a much simpler fusion strategy. For RGB-D, we copy the depth map to 3 channels and concatenate it with RGB along the width dimension; for FOA, we flatten the 4 channels into one long waveform and extract features as if it were monaural audio. This replaces the depth-aware visual encoding and structured FOA spatial modeling while keeping the input modalities unchanged. The results show that this simple matched-input fusion is consistently much worse than both JAEGER variants across localization, grounding, and especially joint reasoning, indicating that the gains do not come merely from richer modalities, but from how JAEGER represents and aligns them. Here, a-o-1 / a-o-2 denote single-source / overlapping-source audio-only DoA median error, v-o IoU / v-o Offset denote visual-only 3D grounding IoU / center offset, and a-v-1 / a-v-2 denote single-speaker / two-speaker audio-visual reasoning accuracy; the six columns follow the same task order as Table 2 in the paper.
> |Model|a-o-1 (°)↓|a-o-2 (°)↓|v-o IoU↑|v-o Offset (m)↓|a-v-1 (%)↑|a-v-2 (%)↑|
> |:-:|:-:|:-:|:-:|:-:|:-:|:-:|
> |SimpleFuse|3.40|21.90|0.30|0.17|67.79|46.59|
> |JAEGER(Classical)|2.95|16.09|0.32|0.16|99.5|98.6|
> |JAEGER(Neural)|2.21|13.13|0.32|0.16|99.5|99.2|
> - **W4: Visual grounding performance is not very impressive.**
>
> Our goal is not to claim standalone visual grounding SOTA. Task visual grounding is used to test whether the model can recover sufficiently accurate 3D object locations for downstream joint audio-visual reasoning. This is stricter than standard 2D grounding, since the model must predict full 3D bounding boxes rather than 2D regions. We therefore report both 3D IoU and Visual Offset. While the 3D IoU is 0.32, the median visual offset is only 0.16 m, indicating accurate metric localization of candidate sound-emitting objects. This is sufficient for downstream AV matching and is not the main bottleneck of the overall reasoning task. We will clarify this framing in the revision.
>
> [1] Shimada K, et al. STARSS23: An Audio-Visual Dataset of Spatial Recordings of Real Scenes with Spatiotemporal Annotations of Sound Events. In Proc. NeurIPS, 2023.

---

> > ### Author Rebuttal · Reviewer_dJWu · 2026-04-02
> >
> > Thank you for the rebuttal. Most of my concerns are resolved.

---

> > > ### Author Response · Authors · 2026-04-08
> > >
> > > Thank you for carefully reading the rebuttal and for the thoughtful follow-up. We appreciate that you found most of your concerns resolved and are grateful that you updated your score accordingly. We also understand your point about relying on the broader consensus and the AC’s judgment for the final decision, and we sincerely appreciate your fair and careful assessment.

---

### Decision · Program_Chairs · 2026-04-30

**Decision:**

Accept (regular)

**Comment:**

The reviewers have achieved agreement on accepting the paper.